# White Matter Lesions Identified by Magnetic Resonance in Women with Migraine: A Volumetric Analysis and Clinical Correlations

**DOI:** 10.3390/diagnostics13040799

**Published:** 2023-02-20

**Authors:** Natália de Oliveira Silva, Nicoly Machado Maciel, Júlio César Nather, Jr., Gabriela Ferreira Carvalho, Carina Ferreira Pinheiro, Marcelo Eduardo Bigal, Antônio Carlos dos Santos, Debora Bevilaqua-Grossi, Fabiola Dach

**Affiliations:** 1Department of Neurosciences and Behavioral Sciences, Ribeirão Preto Medical School, University of São Paulo, Ribeirão Preto 14015-010, SP, Brazil; 2Department of Health Sciences, Ribeirão Preto Medical School, University of São Paulo, Ribeirão Preto 14015-010, SP, Brazil; 3Department of Internal Medicine, Ribeirão Preto Medical School, University of São Paulo, Ribeirão Preto 14015-010, SP, Brazil; 4Department of Physiotherapy, Institute of Health Sciences, University of Luebeck, 23562 Luebeck, Germany; 5President and CEO, Ventus Therapeutics, Boston, MA 02453, USA

**Keywords:** migraine, white matter lesions, magnetic resonance, volumetry, voxel-based morphometry

## Abstract

Background: Repeated migraine attacks and aura could independently cause structural changes in the central nervous system. Our research aims to study the correlation of migraine type, attack frequency, and other clinical variables with the presence, volume and localization of white matter lesions (WML), in a controlled study. Methods: Sixty volunteers from a tertiary headache center were selected and divided equally into four groups: episodic migraine without aura (MoA), episodic migraine with aura (MA), chronic migraine (CM) and controls (CG). Voxel-based morphometry techniques were used to analyze WML. Results: There were no differences in WML variables between groups. There was a positive correlation between age and the number and total volume of WMLs, which persisted in the comparison categorized by size and brain lobe. Disease duration was positively correlated with the number and total volume of WML, and when controlled by age, the correlation maintained significance only for the insular lobe. Aura frequency was associated with frontal and temporal lobe WMLs. There was no statistically significant correlation between WML and other clinical variables. Conclusion: Migraine overall is not a risk factor for WML. Aura frequency is, however, associated with temporal WML. Disease duration, in adjusted analyses that account for age, is associated with insular WML.

## 1. Introduction

Migraine is a common neurological condition, but it involves complex mechanisms. Activation of the trigeminovascular system is the pillar of the migraine attack that culminates in neurogenic inflammation and the release of neuropeptides such as CGRP, and PACAP 38. Macroscopically, vasodilation is observed; while microscopically, the activation of specific serotonergic and dopaminergic receptors is responsible for most of the symptoms [1].

It has been suggested that the recurrence of migraine attacks throughout life, especially in the chronic form, results in structural alterations in the brain, a biologically plausible possibility that could be explained by the electrical, biochemical and vascular phenomena that involve the pathophysiology of migraine [2,3,4].

This hypothesis has been first investigated with CT scans which reported cerebral edema, atrophy and ischemia [5]. With the advent of MRI and high-resolution devices, the most commonly reported structural changes were white matter lesions (WML) [6].

WMLs are associated with axonal myelin loss, neuronal rarefaction and gliosis. The pathophysiology of WML formation is multifactorial. The vascular mechanism suggests that the cumulative effect of hypoperfusion periods during migraine attacks could cause microvascular ischemia [7,8]. In addition, it has been suggested that migraine could be associated with a chronic proinflammatory and procoagulant state that contributes to endothelial dysfunction. Neurogenic inflammation promotes excitatory toxicity through the release of inflammatory cytokines and vasoactive neuropeptides. As a cellular mechanism, apoptosis induced by intracellular calcium increase and intra-axonal mitochondrial alteration has been suggested [9]. 

Although the clinical significance of these lesions is still uncertain, it was hypothesized that WMLs may be biomarkers for migraine [10]. Historically, migraine is associated with a higher risk of cardiovascular and cerebrovascular events, especially in older patients with associated vascular diseases or in young women, smokers, and under oral hormonal contraceptive use. In this context, it is questioned whether the occurrence of LSBs could predict which patients would be more prone to such outcomes [11]. 

Advancing on the elucidation of this question, whether migraine is a risk factor for brain damage is, therefore, clinically relevant. Volumetric MRI techniques allow better assessment of WMLs and may provide a different comprehension of structural aspects related to the migraine pathophysiology. The identification of migraineurs’ brain structural alterations through non-invasive neuroimaging methods and the use of accessible and easy-to-use software brings new perspectives on the assessment of the impact of this condition in central nervous system macroarchitecture [12,13].

The present study compares quantitative variables of WMLs, such as number, volume, localization and size categories, between four groups: episodic migraine with aura, episodic migraine without aura, chronic migraine and control. In addition, this study intends to identify which clinical variables most influence WML formation and to establish correlations between the clinical and volumetric variables.

## 2. Materials and Methods

This is a cross-sectional study performed in a tertiary hospital. Sixty female volunteers were included, aged between 18 and 55 years old, who were equally allocated into four groups: episodic migraine without aura (MoA), episodic migraine with aura (MA), chronic migraine (CM) and controls (CG). Patients were recruited from the headache outpatient clinic or invited by e-mail. Healthy controls were recruited from patients’ relatives and hospital staff.

The diagnoses were made by headache specialists and based on ICHD-3 beta criteria. General and neurological examinations were normal. The excluding criteria were: the presence of cerebrovascular risk factors, such as Diabetes mellitus, systemic arterial hypertension, dyslipidemia, history of acute myocardial infarction, ischemic or hemorrhagic stroke and obstructive vasculopathy; neurological degenerative diseases; history of neurosurgery or brain trauma; alcoholism or smoking in the last 10 years; BMI > 30 kg/m^2^; pregnancy; the presence of another primary or secondary headache; claustrophobia; the presence of metallic devices that could interfere with the acquisition of magnetic resonance images. 

Structured questionnaires and clinical interviews were used to clinically characterize the sample by the following variables: age, number of days with headache in the last month, number of days with aura in the last month, average pain intensity in the last month accessed by numerical visual scale, disease duration calculated in the number of years since migraine diagnosis, and the use of prophylactic drugs for migraine. Although the clinical data were collected taking into account the last month, clinical characteristics should be stable in the last three months.

### 2.1. Images Acquisition

Volunteers underwent brain MRI imaging in a Philips 3 Tesla device, in a 32-channel phased-array coil. Image acquisition protocol included the following sequences: 2D axial T2-weighted images (turbo spin-echo); T2-weighted images with fluid suppression (Fluid attenuated Inversion Recovery—FLAIR), acquired in the sagittal plane and reconstructed in the orthogonal planes, with isotropic voxel; T1-weighted, 3D gradient-echo (MPRAGE), high-contrast, isotropic voxel images acquired in the sagittal plane and reconstructed in the three orthogonal planes. 

The following technical parameters were used for VBM (voxel-based morphometry): voxel size 1 × 1 × 1 mm^3^; field of 256 mm view and 256 mm and 256 mm matrix; 176 sagittal slices, slice thickness 1 mm; recovery time 2530 ms; echo time 1.64/3.5 ms; inversion time 1200 ms; 651 Hz/px bandwidth for all echoes and 7° excitation angle (non-selective). 

### 2.2. Image Processing

Volumetric analysis used Freesurfer (version 4.05) and 3D-Slicer (Version 3.6) software, both open source. 

Freesurfer is an automated instrument for the reconstruction and segmentation of brain structures and is based on a structural atlas. It promotes the alignment of images in Talairach space to obtain a voxel-to-voxel correspondence and builds an electronic archive of the regions under study. After that, maps are created using spatial intensity gradients between tissue classes without resolution restriction of voxels from the original data; therefore, they can detect submillimeter differences between the studied groups. 

The 3D-Slicer was the platform used for the analysis (including recording and interactive segmentation) and visualization of medical images (covering the measurement of the volume of anatomical structures). 

### 2.3. WMLs Identification and Analysis

WMLs were defined as hyperintense areas on T2 and FLAIR sequences, without corresponding hypointensity on T1 sequences. Only WMLs visible in at least two consecutive 3 mm thick axial slices were included in the study. 

This specific thickness was chosen to minimize errors in the WML identification since thinner thickness could result in mistaken inclusion of perivascular spaces and the higher thickness could exclude lesions whose volume would be statistically significant. 

WMLs were first identified by a neurologist experienced in neuroradiology and familiar with the software. Then, each exam was reviewed by two independent blinded neuroradiologists. The presence and size of WMLs variables were a consensus between them. Segmentation was performed manually, voxel-by-voxel, from the volumetric maps in the high-resolution FLAIR sequence. 

The data obtained from the neuroimaging used were: WMLs absolute number, WMLs total volume and WMLs mean volume. Volume lesions were categorized by size percentiles into small (<0.034 mL), medium (0.034–0.059 mL), or large (>0.059 mL). In terms of location, WMLs were classified by cerebral lobe (frontal, temporal, insular, parietal or occipital), without considering laterality. 

Relative variables, expressed in percentage ratios, were constructed to equalize the proportion of lobes sizes and the amount of white matter which varies between individuals. 

### 2.4. Statistical Analysis

The Mann–Whitney test was used to compare two groups concerning a quantitative variable; while the Kruskal–Wallis test was used to compare more than two independent groups. Dunn’s test was used for multiple comparisons. The chi-square test or Fisher’s exact test was used to assess the association between two qualitative variables. 

Spearman’s correlation coefficient was used to quantify the correlation between two quantitative or ordinal variables. Mukaka’s reference values were used to interpret correlation strength.

## 3. Results

The clinical characteristics of our sample are shown in Table 1. 

The low global prevalence of WMLs in our sample is noteworthy, almost half of the volunteers had no lesions. The WMLs were mostly small size in all groups. No infratentorial lesions were identified. The WMLs were exclusively supratentorial, predominantly in anterior cerebral blood circulation territories. It is important to observe that the value of the median volume of WML was 0.00 because most patients with chronic migraine did not have WML.

The statistical analyses showed no differences in WML volumetric variables between the four groups as seen in Table 2. The groups remained similar in WML comparison categorized by size and location and also in the dichotomous assessment (presence vs. absence) of lesions. 

There was a higher prevalence of WMLs in the frontal and parietal lobes in all groups, which is expected since these lobes are the most voluminous. In order to minimize this bias, relative volume variables were calculated: the volume of WMLs by lobe in relation to the total volume of white matter in that same lobe. These data provide a better idea of the amount of tissue affected by lesions in relation to the size of the structure. Such analysis also showed a higher relative volume of WMLs in the frontal and parietal lobes. However, when comparing brain lobes by pair analysis, a significant difference was found only when the frontal and parietal lobes were compared to the occipital lobe, whose occurrence of WMLs was rare in our sample. When comparing the relative volumes of WMLs between the groups, there were also no statistically significant differences. 

Finally, we tested correlations between clinical and volumetric variables, which are demonstrated in Table 3 and Table 4. A weak positive correlation was found between age and the number of WMLs (*p* = 0.001; rho 0.43), as well as between age and WMLs total volume (*p* = 0.001). This correlation persisted when WMLs were categorized by size and location. In correlation tests between age and relative volume variables, the same behavior was observed, except for the occipital lobe, which can be interpreted as an outlier because of the very low occipital WMLs prevalence in our sample. 

Disease duration was positively correlated with the following brain volumetric parameters: WML total volume (*p* = 0.015; r = 0.365), WML mean volume (*p* = 0.043; r = 0.307), WML total number (*p* = 0.015; r = 0.365), number of large WMLs (*p* = 0.027; r = 0.334), the volume of large WMLs (*p* = 0.032; r = 0.324), and the ratio of total WML volume divided by cerebral white matter total volume (*p* = 0.013; r = 0.373); however, all correlations found were of weak. Since statistics revealed that disease duration is moderately influenced by age (*p* < 0.0001; rho = 0.554), when disease duration was controlled by age, the unique correlation observed was with WML volume in the insular lobe (*p* = 0.039).

The frequency of aura was studied in the MA and CM groups. Positive correlations were found with the number of frontal lobe WMLs (*p* = 0.004), the number of temporal lobe WMLs (*p* = 0.037), the volume of temporal lobe WMLs (*p* = 0.037), the relative volume of temporal WMLs (*p* = 0.045) and the number of small WMLs (*p* = 0.038). 

In relation to attack frequency, headache intensity and use of prophylactics, no correlations were found with WMLs variables.

## 4. Discussion

This study showed no differences between the MoA, MA, CM, and GC groups in our sample regarding the total number, volume and size of WMLs, and the number, volume and size of WMLs separated by cerebral lobes. 

The CAMERA-1 study found no differences in the global analysis between migraineurs and the control group, as in our study [14]. However, in the subgroup analysis of young women with migraine, the number of deep WMLs was higher when compared to the control group, this association was independent of the type of migraine. 

The same sample was evaluated in the CAMERA-2 study, after 9 years of follow-up, and demonstrated that the volume of WMLs is greater in the group of women with migraine when compared to the control group; but in the global analysis, this difference was not statistically relevant [15]. 

The ARIC MRI study showed an association between migraine without aura and increased risk for more severe levels of WML (OR 1.87) [16]. However, when the groups were directly compared in a separate model, the difference was not statistically significant. 

The unique metanalysis emphasizes the heterogeneity of the studies, whose prevalence of WMLs in migraine patients is extremely variable (4 to 59%) [17]. Despite concluding that migraine with aura is associated with a higher risk of WMLs in women, it emphasizes that further studies are needed. 

Our study results are in agreement with a last population-based study, which compared WMLs in twins with migraine with aura, migraine without aura and controls [18]. The study concluded that there were no differences between groups. 

### 4.1. WMLs Locations 

In our study, we evaluated the location of WMLs by cerebral lobe. The frontal and parietal lobes had the highest number and highest volume of WMLs; however, as they are the largest lobes of the brain, these lesions are more likely to occur in these lobes. In relative variables analyses, the difference persisted in the comparison by pairs of these lobes with the occipital lobe, whose incidence of WMLs was minimal in the sample. Despite the higher prevalence of frontal WML, a previous study with medication-overuse-headache showed a negative correlation between WML and impaired cognitive function [19]. In the literature, few studies performed a volumetric analysis of WMLs per cerebral lobe [8,9]. A study that compared WMLs in migraineurs and multiple sclerosis patients found predominance, in quantity, in the frontal and parietal lobes, but no difference was found in the comparison of WML volumes between the lobes. There was a higher prevalence of WMLs in supratentorial and anterior circulation [8]. Furthermore, it reports that posterior circulation WMLs were more common in the MA group. The results were similar in our sample, specifically, there was a single patient with occipital WMLs, belonging to the CM group, who reported attacks with aura on 15 days of the last month.

Another study showed a predominance of lesions in the frontal and parietal lobes and a higher prevalence of deep WMLs; however, it did not analyze the vascular territory involved [9].

CAMERA-1 study found a higher risk of deep and supratentorial WMLs in migraine patients compared to controls but did not analyze the location of lesions in terms of arterial circulation or cerebral lobes [14]. Kruit et al. described a prevalence of 4.4% of infratentorial WMLs, they were almost always associated with the concomitant occurrence of supratentorial WMLs [20].

We did not identify infratentorial WMLs in our sample. Our results are very similar to Senevirate et al. study, in which no infratentorial WMLs were seen in the cohort of 44 patients [21]. Uggetti et al. also did not identify WMLs in posterior circulation territory in a sample exclusively of migraine with aura [22].

### 4.2. Correlation with Clinical Variables 

Regarding the clinical characteristics of our sample, we found weak correlations with age, disease duration and aura frequency. 

Age is an independent risk factor for the formation of WMLs. The WMLs prevalence increases with age, even in individuals who do not have migraine and may be related to the higher prevalence of cardiovascular risk factors in older ages [23,24,25,26].

In our study, age was the clinical characteristic most related to WML occurrence. It was positively correlated with the total number and total volume of WML. Such correlation was maintained when analyzing WML occurrence per cerebral lobe, except for the occipital lobe. This result was similar when testing the correlation of age and the relative variables of WMLs. The correlation was also positive between age and all WMLs sizes.

A literature review points out that age is an independent risk factor for both periventricular and deep WML formation in migraineur women [27].

Cross-sectional studies did not find a significant correlation between WML variables with age. A VBM study showed that migraineurs with WMLs were significantly older than lesion-free ones, but it does not influence brain global volumes, or cortical thickness [28]. 

Follow-up studies may represent a more reliable way of assessing the influence of age on the formation of WMLs because they could assess the progression of WMLs. 

CAMERA-2 showed an increase in deep WLMs mean volume after 9 years of only in the group of women with migraine without aura [15]. ARIC-MRI, whose follow-up ranged from 8 to 12 years, concluded that migraine is associated with WMLs only in cross-sectional analyses, but not with the WMLs progression over time [16]. A smaller study, with a 3-year follow-up, showed an increase in WMLs number compared to baseline; however, neither growth nor reduction of WMLs was related to age [9].

Disease duration reflects the years of disease evolution since the diagnosis; therefore, individuals with longer disease duration had their brains exposed for a longer time and could be more prone to WML formation. 

We found a positive correlation between disease duration and the following WML variables: total number, total volume, mean volume and relative volume, also with the number and volume of large WMLs. When controlled by age, we found a weak positive correlation between disease duration and insular WML volume, which was maintained in the assessment of the relative volume of insular WMLs.

Insula is involved in cognitive-behavioral and emotional aspects of pain processing; receives nociceptive afferences from the trigeminovascular system and sends efferences to the autonomic nervous system [29]. 

A functional MRI study in migraine demonstrated increased insular activity in the interictal period, which supports the hypothesis that repeated migraine attacks modify insular function throughout life [30]. Other studies revealed increased connectivity of the anterior insula to the dorsal pons and primary visual and auditory cortices, while there was reduced connectivity of the posterior insula to the thalamus and various cortical regions [29,30].

The Danish population-based study with twins did not find relevant correlations of volumetric variables with disease duration. However, participants were on average 15 years older than our volunteers and cardiovascular risk factors were not excluded [18]. 

Another study reported that migraineurs with WMLs were older compared to migraineurs without WMLs and controls. In that study, the disease duration, attack frequency and aura frequency were significantly higher among those with WML [28]. 

Zhang et al. found no difference in lesion burden between patients with migraine with aura and controls, and there was no correlation between WMLs and migraine duration or with the frequency of attacks [31]. Similar results were published by Galli et al. in a study with 90 patients with visual, sensory or aphasic aura [32].

A prospective study showed that migraineurs with WML at baseline were older and had more years of a migraine diagnosis. At follow-up, the group that had no improvement in symptoms had a higher WML burden [29]. 

The association of WMLs with age and disease duration raises questions about the different vascular mechanisms in WML formation. In younger patients, the main influencing factor may be the attack frequency, while in older patients, the disease duration may be more relevant, because it is expected that attack frequency decreases with age and concomitantly there is an increase in cardiovascular diseases [15]. 

The correlation between aura frequency and WML variables were tested in the MA and CM groups and a positive correlation was found for the following parameters: WML total number, and WML number in frontal and temporal lobes, temporal lobe WML volume and the ratio of temporal lobe WML volume over temporal lobe volume. Above all, the correlations with the temporal lobe stand out, which remained relevant for the relative variables of this lobe. 

The temporal lobe is a multisensory integration zone and participates in nociceptive and visual processing; therefore, it may be involved in the visual aura mechanisms. VBM studies demonstrated cortical thickness reduction in the temporal lobe of migraine patients, while functional studies have revealed temporal pole hyperexcitability in these patients [33]. 

Although our sample was not categorized by aura type, we expected to find more WMLs in the occipital lobe, since the visual aura represents the most prevalent type of aura [34]. 

Bashir et al. concluded that the aura type does not influence the location and distribution of WMLs [17]. No population study, to the present date, has evaluated the aura frequency [14]. Previously, it was hypothesized that cortical spreading depression would be capable of triggering neurogenic inflammation and subsequent vascular phenomena, which were related to WML formation; however, our study did not confirm this hypothesis. 

The attack frequency could influence the WML formation given the cumulative effect of repeated attacks on the brain. However, this hypothesis has been refuted by several studies that, like ours, did not show a correlation between attack frequency and WML variables [17,18,34].

Contrary to these studies, CAMERA-1 showed a higher relative risk of WMLs for patients with an attack frequency of more than one episode per month, which is a very low cut-off value [14]. 

A small prospective study showed that patients with lower attack frequency at baseline were more likely to have their WMLs disappear [9]. This finding raised questions about the dynamic nature of WMLs, as small lesions are more likely to decrease in size, while large lesions are more likely to increase; in addition, new lesions were noted in some patients and the disappearance of previous lesions in other patients. Therefore, the question of if the formation of WMLs is a reversible or progressive process remains unclear. 

The possible reversibility of WMLs raises reflections on the role of antimigraine therapy as a modifier of disease course and in the prevention of WMLs formation. In our sample, there was no correlation between the use of prophylactics and WML variables. Only one study has evaluated prophylactic therapy during a 3-month period and has not found differences in WML variables [35]. The use of abortive therapy was not evaluated in our study; previous research showed that the use of opioids increases the risk of cardiovascular or cerebrovascular events and triptan use diminishes that risk [36]. 

Finally, pain intensity also showed no correlation with WMLs parameters, which is consistent with the literature [15,17]. 

### 4.3. Strength

To our knowledge, this is the first study to perform volumetric WMLs analysis simultaneously in four distinct equally distributed groups: MA, MoA, CM, and CG. Additionally, this is the first study performed in Latin America on WMLs. 

The automatized software utilized is precise and capable to identify minimum differences between the tecidual areas. The identification of WML had a high accuracy because it was performed by a consensus of three independent professionals.

The volumetric analysis presented was extensive and evaluated not only WML number and volume but also characterized them by location and size percentiles. Furthermore, relative variables were included, an analysis also unprecedented in the literature. 

### 4.4. Limitations

Our total sample size may be considered small; however, it is well-balanced between groups.

The low overall WML prevalence, the absence of infratentorial lesions, and the low occurrence of WMLs and ILS in territories of posterior arterial circulation hampered the analysis of these anatomical sites.

Regarding the WML location, we used the division by lobes and did not evaluate the distribution of WMLs as periventricular, deep, subcortical or pericallosal, which could reveal new information. 

The assessment of the use and abuse of abortive medications for migraine attacks could also contribute to understanding the role of antimigraine therapies on WML formation. Additional analysis of the type of aura could provide further clarification. However, since the visual aura is the most prevalent in migraineurs, our study showed that, apparently, it was not related to the occurrence of WMLs in the occipital lobe.

## 5. Conclusions

In general, neither the frequency of attacks nor the presence or frequency of aura was correlated with the highest number and highest volume of WMLs. In subgroup analysis, aura frequency was correlated especially with temporal lobe WMLs. Regarding the clinical variables, it is noteworthy that the time of disease corrected by age was relevant for insular WML. These findings may indicate that these anatomical sites play a role in the pathophysiology of migraine.

No infratentorial WMLs were found in our sample. Among the supratentorial WML, there was a predominance in the anterior circulation. This refutes the previous hypothesis that posterior regions would be more prone to the formation of LSB because they are more susceptible to the hemodynamic changes of migraine attacks [17]. 

The relationship between migraine and WMLs remains unclear, as well as the clinical significance of these lesions. It is not possible to establish whether migraine is a risk factor for the formation of WMLs, nor whether WMLs are able to predict complications. 

New studies that correlate structural and functional neuroimaging as well as the accomplishment of cohorts with longer follow-up can contribute to understanding the role of WMLs as an imaging marker in migraine.

## Figures and Tables

**Table 1 diagnostics-13-00799-t001:** Sample clinical characteristics.

	Group	Media	Median	DP	*p*-Value
Age (years)	MoA	33	32	0.78	0.964
	MA	33.86	35.50	7.70	
	CM	32.53	29.00	9.36	
	CG	32.87	32.00	9.68	
Attack frequency *	MoA	6.60	6.00	2.97	<0.0001
	MA	7.50	7.50	2.56	
	CM	23.27	25.00	5.50	
Aura frequency **	MA	4.07	3.50	2.37	<0.0001
	CM	1.53	0.00	3.96	
Pain intensity (0–10)	MoA	7.53	8.00	0.83	0.585
	MA	7.79	8.00	1.58	
	CM	7.80	8.00	1.61	
Disease Duration (years)	MoA	17.27	18.00	8.11	0.548
	MA	17.50	15.00	8.99	
	CM	20.00	20.00	7.99	

* headache days/month; ** aura days/month. **Abbreviations**: MoA—migraine without aura; MA—migraine with aura; CM—chronic migraine; CG—control group. ***p*-value**: calculated based on comparison between the groups.

**Table 2 diagnostics-13-00799-t002:** Volumetric variables compared between groups.

		Mean	Median	SD	Min	Max	*p*-Value
WML total volume	MoA	163.27	19.04	316.33	0.00	1118.80	0.540
	MA	115.61	28.47	200.69	0.00	692.19	
	CM	45.62	0.00	81.63	0.00	253.74	
	CG	105.17	24.98	150.36	0.00	518.56	
WML mean volume	MoA	26.04	9.76	31.45	0.00	103.88	0.691
	MA	13.70	12.22	15.22	0.00	48.03	
	CM	16.80	0.00	29.29	0.00	104.68	
	CG	19.20	14.18	26.31	0.00	104.68	
WML total number	MoA	4.07	1.00	9.25	0.00	36.00	0.396
	MA	4.79	2.00	8.05	0.00	29.00	
	CM	1.53	0.00	3.58	0.00	14.00	
	CG	5.00	1.00	7.31	0.00	19.00	
Frontal WML number	MoA	2.13	0.00	5.89	0.00	23.00	0.409
	MA	2.79	1.00	4.44	0.00	13.00	
	CM	0.93	0.00	1.94	0.00	7.00	
	CG	2.40	0.00	3.64	0.00	10.00	
Frontal WML volume	MoA	84.45	0.00	222.92	0.00	869.63	0.510
	MA	65.34	11.63	101.71	0.00	307.05	
	CM	29.84	0.00	60.67	0.00	209.37	
	CG	42.98	0.00	62.78	0.00	187.59	
Parietal WML number	MoA	1.33	0.00	2.58	0.00	8.00	0.356
	MA	1.64	0.00	3.73	0.00	14.00	
	CM	0.40	0.00	1.06	0.00	4.00	
	CG	2.00	0.00	3.18	0.00	10.00	
Parietal WML volume	MoA	67.77	0.00	143.46	0.00	492.74	0.431
	MA	36.63	0.00	81.82	0.00	302.35	
	CM	12.68	0.00	32.40	0.00	117.69	
	CG	53.42	0.00	89.73	0.00	293.91	
Temporal WML number	MoA	0.33	0.00	0.82	0.00	3.00	0.665
	MA	0.21	0.00	0.43	0.00	1.00	
	CM	0.07	0.00	0.26	0.00	1.00	
	CG	0.33	0.00	0.72	0.00	2.00	
Temporal WML volume	MoA	5.20	0.00	12.04	0.00	39.65	0.696
	MA	4.19	0.00	9.07	0.00	29.93	
	CM	1.37	0.00	5.30	0.00	20.51	
	CG	4.51	0.00	10.48	0.00	37.04	
Insular WML number	MoA	0.27	0.00	0.70	0.00	2.00	0.526
	MA	0.14	0.00	0.36	0.00	1.00	
	CM	0.00	0.00	0.00	0.00	0.00	
	CG	0.27	0.00	0.80	0.00	3.00	
Insular WML volume	MoA	5.85	0.00	15.59	0.00	49.39	0.524
	MA	9.46	0.00	30.87	0.00	115.56	
	CM	0.00	0.00	0.00	0.00	0.00	
	CG	4.27	0.00	12.30	0.00	45.04	
Occiptal WML number	MoA	0.00	0.00	0.00	0.00	0.00	0.402
	MA	0.00	0.00	0.00	0.00	0.00	
	CM	0.13	0.00	0.52	0.00	2.00	
	CG	0.00	0.00	0.00	0.00	0.00	
Occiptal WML volume	MoA	0.00	0.00	0.00	0.00	0.00	0.402
	MA	0.00	0.00	0.00	0.00	0.00	
	CM	1.73	0.00	6.69	0.00	25.92	
	CG	0.00	0.00	0.00	0.00	0.00	

**Abbreviations**: MoA—migraine without aura; MA—migraine with aura; CM—chronic migraine; CG—control group; WML—Whitte matter lesions. ***p*-value**: calculated based on comparison between the groups.

**Table 3 diagnostics-13-00799-t003:** Correlations between volumetric parameters by lobes and clinical features (Spearman’s correlation—r-value).

	Frontal			Temporal			Insular			Parietal			Occiptal		
	Number of WML	WML Absolute Volume	WML Relative Volume	Number of WML	WML Absolute Volume	WML Relative Volume	Number of WML	WML Absolute Volume	WML Relative Volume	Number of WML	WML Absolute Volume	WML Relative Volume	Number of WML	WML Absolute Volume	WML Relative Volume
Age ^a^	0.421 **	0.388 **	0.399 **	0.384 **	0.398 **	0.399 **	0.260 *	0.259 *	0.259 *	0.376 *	0.362 *	0.369 *	0.169	0.170	0.169
Disease Duration ^a^	0.294	0.262	0.278	0.188	0.214	0.214	0.048	0.025	0.025	0.260	0.265	0.277	0.247	0.247	0.247
Disease duration ^a,b^	0.013	0.014	0.144	0.085	0.029	0.024	0.133	0.316 *	0.317 *	0.178	0.117	0.104	0.240	0.240	0.240
Frequency attack ^c^	0.003	0.037	0.030	0.008	0.017	0.017	0.051	0.045	0.045	0.015	0.099	0.099	0.242	0.242	0.242
Aura Frequency ^c^	0.377 *	0.325	0.323	0.388 *	0.375 *	0.375 *	0.327	0.328	0.328	0.257	0.262	0.263	0.326	0.326	0.326
Headache Intensity ^d^	0.225	0.189	0.19	0.051	0.031	0.842	0.036	0.013	0.013	0.239	0.242	0.236	0.254	0.254	0.254

^a^ years; ^b^ controlled by age; ^c^ days per month; ^d^ Numeric Visual Scale. * *p* < 0.05; ** *p* < 0.01.

**Table 4 diagnostics-13-00799-t004:** Correlations between volumetric parameters (whole brain) and clinical features.

	Number of WML	WML Absolute Volume	WML Relative Volume
Age	0.430 **	0.421 **	0.421 **
Disease Duration ^a^	0.365 *	0.365 *	0.373 *
Disease duration (controlled by age) ^a,b^	0.140	0.158	0.147
Frequency attack ^c^	0.078	0.056	0.054
Aura Frequency ^c^	0.321	0.263	0.263
Headache Intensity ^d^	0.162	0.164	0.153

^a^ years; ^b^ controlled by age; ^c^ days per month; ^d^ Numeric Visual Scale. * *p* < 0.05; ** *p* < 0.01.

## Data Availability

The data presented in this study are available on request from the corresponding author. The data are not publicly available due to privacy or ethical restrictions.

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
