# Peer review of "White Matter Lesions Identified by Magnetic Resonance in Women with Migraine: A Volumetric Analysis and Clinical Correlations"

_diagnostics, 2023, doi:10.3390/diagnostics13040799_

Round 1
Reviewer 1 Report
The research the authors submit for bla referees' evaluation concerns an area of headache medicine, and in particular migraine, that has been explored extensively over the years, especially in the past decade. Indeed, there have been many studies on the diagnostic, prognostic or predictive significance of comorbidity in migraine with aura and White Matter Lesions. Certainly the cardiovascular risk with MWA is established, but there are a number of studies in the literature that are not always concordant and with different points of view.
On the other hand, this is an eventuality that confronts the clinician very frequently and a review may be useful.
I would suggest implementing the manuscript with the following papers:
PMID: 23565964
PMID: 34168492
https://doi.org/10.1007/s42399-020-00576-7
PMID: 34645382
https://doi.org/10.1186/s10194-019-0959-2
Author Response
Suggestions for Authors – Reviewer 1
“The research the authors submit for bla referees' evaluation concerns an area of headache medicine, and in particular migraine, that has been explored extensively over the years, especially in the past decade.”
Reply: In fact, the study of T2/FLAIR hyperintensities in migraine patients has been extensively explored in recent years; however, we consider that it was not exhaustive, especially in terms of voxel-based morphometry.
“Indeed, there have been many studies on the diagnostic, prognostic or predictive significance of comorbidity in migraine with aura and White Matter Lesions. Certainly the cardiovascular risk with MWA is established, but there are a number of studies in the literature that are not always concordant and with different points of view. On the other hand, this is an eventuality that confronts the clinician very frequently and a review may be useful.”
Reply: Our study is the only one that simultaneously evaluates four groups: episodic migraine without aura, episodic migraine with aura, chronic migraine, and healthy controls. As a differential, we also performed relative volume calculations, to reduce biases and assess how significant the WML volume is in relation to global volumes and brain lobe volumes. Finally, it is the only research on the subject carried out in Latin America
“ I would suggest implementing the manuscript with the following papers: 1) PMID: 23565964; 2) PMID: 34168492; 3) https://doi.org/10.1007/s42399-020-00576-7; 4) PMID: 34645382; 5) https://doi.org/10.1186/s10194-019-0959-2.”
Reply: We appreciated the suggestion related to the insertion of some important reference. We had added them respectively in pages: 02, 10, 10, 12 e 11.

Reviewer 2 Report
The work presented to me for review is undoubtedly very important and brings new knowledge in the field of neuroimaging changes in migraine. For many years there has been a lot of controversy on this topic which creates many diagnostic and therapeutic difficulties.
The authors did not show a correlation of lesion volume with disease duration. The study is carried out on a large number of patients, enough to make adequate statistics. It is written according to a typical scheme, the results and conclusions are presented in a clear manner.
It is worth writing two sentences in the introduction about the pathogenesis of migraine and the neurotransmitter localization of potential disorders based on the article: https://pubmed.ncbi.nlm.nih.gov/34073933/
Author Response
“The work presented to me for review is undoubtedly very important and brings new knowledge in the field of neuroimaging changes in migraine. For many years there has been a lot of controversy on this topic which creates many diagnostic and therapeutic difficulties.”
Reply: We are very thankful for your analyses and appreciate your comments on the review. It is indeed a controversial theme, in terms of clinical significance of white matter lesions and its value as a biomarker of possible cardiovascular and cerebrovascular complications.
“The authors did not show a correlation of lesion volume with disease duration. The study is carried out on a large number of patients, enough to make adequate statistics. It is written according to a typical scheme, the results and conclusions are presented in a clear manner.”
Reply: Regarding the correlation between WML volumes and disease duration, the values are expressed in table 3. These results were pointed out in the text on page 3, second paragraph: we pointed out that disease duration was positively correlated with total, mean and relative volumes of WML, also with volume of large WML (defined as >0,059 ml). When disease duration was adjusted by age, we found a weak positive correlation only with insular WML volumes.
“It is worth writing two sentences in the introduction about the pathogenesis of migraine and the neurotransmitter localization of potential disorders based on the article: https://pubmed.ncbi.nlm.nih.gov/34073933/”
Reply: As suggested, we included information about migraine pathogenesis and neurotransmitters in the manuscript introduction, pages 01 and 02, based on the article suggested above.

Reviewer 3 Report
The authors have studied a very interesting and intriguing subject, but nevertheless with not many scientific advances.
The overall idea and drafting of the study are very well implemented.
Nevertheless, there are some issues that need attention.
Please see the attached pdf with corrections.
The sample size is very small, especially when taking into account the large span of age of the patients that are included in the study.
Please make appropriate corrections.

Author Response
We appreciated all the observation and criticisms. We hope that all of them were answered
bellow:
Page 2, line 84:
ICHD-3 beta criteria were the operative version at the time of the first patients’ inclusion, that
is, the ICHD-3 definitive version was not released yet. However, as there were no differences
between the beta and definitive versions of ICHD 3 regarding the migraine criteria, we changed
ICH3 beta version to ICHD 3, 2018.
Page 2, line 85:
We appreciated this observation. It is really important to specify the cerebrovascular risk factors
taken into account as exclusion criteria. They were diabetes mellitus, systemic arterial
hypertension, dyslipidemia, history of acute myocardial infarction, ischemic or hemorrhagic
stroke and obstructive vasculopathy. They were added on the last paragraph of page 2
Page 3, line 2:
We considered the last month to collect information about some clinical variables, such as
frequency, intensity, and duration of attacks in order to reduce possible recall bias since
migraine varies in clinical characteristics throughout life. However, patients should be clinically
stable in the last 3 months, which is the prerogative of ICHD 3 criteria of migraine subtypes
(episodic x chronic). Summing up, the clinical data was collected taking into account the last
month, but clinical characteristics should be stable in the last three months. We clarify this point
in the manuscript in the first paragraph, on page 3.
Page 4, table 1:
Thank you for this observation. We added the missing information in the legend.
Page 4, table 2:
We added the missing information in the legend.
The value of the median volume of WML was 0.00 because most patients with chronic migraine
do not have WML. This information is explained in page 4, line 164
We will provide the raw data on a separated file. Please specify which data is desirable.
Page 9, line 206:
Thank you for this observation. We added each p-value.
Page 9, line 193:
Sorry for this mistake. Chronic migraine abbreviation was corrected to CM throughout the text.
Additional criticism: “The sample size is very small, especially when taking into account the
large span of age of the patients that are included in the study.”
Migraine is a disease that affects individuals since early childhood to old age, so the span of age
of migraine is really large. However, in order to increase the chance to find differences between
groups of migraine and controls, we restrict the analyzes to adult women during menacme.
Regarding the sample size, cross-sectional studies with WML varies a lot, but studies usually include only two or three groups; once ours included four different groups, we consider the sample satisfactory for extensive volumetric analyses.

Round 2
Reviewer 3 Report
Dear authors,
Thank you for taking into consideration my comments.
Please provide raw data in a separate file (supplementary) about the volumetric variables (WML) of patients.
Author Response
The raw data, as asked, was upload in the link below. Please see the attachment.
Because is a large table of data, when it was converted to pdf, the archive result in many pages, but the order of the subjects is the same in the subsequent pages.
If you have any doubts, do not hesitate to contact us.
Thank you again for your review.
